# Three-Dimensional Hollow Tubular Structure of Rocket

## **Chemical Depletion**

3

2

Chunyu Deng<sup>1,2</sup>, Xiangxiang Yan<sup>1,2,3\*</sup>, Tao Yu<sup>1,2,3</sup>, Chunliang Xia<sup>1,2,3</sup>, and Yifan Qi<sup>2,3</sup>

- Hubei Subsurface Multi-scale Imaging Key Laboratory, School of Geophysics and Geomatics, China
- University of Geosciences, Wuhan, 430074, China
- <sup>2</sup>Hubei Key Laboratory of Planetary Geology and Deep-Space Exploration, School of Earth Sciences,
- China University of Geosciences, Wuhan, 430074, China
- 3Key Laboratory of Geological Survey and Evaluation of Ministry of Education, China University of
- Geosciences, Wuhan, 430074, China
- Correspondence to: Xiangxiang Yan (yanxxcug@foxmail.com)

## 13 Key Points:

- (1) The vertical structure of rocket-exhausted ionospheric electron density depletion was
- captured by COSMIC-1 radio occultation data.
- (2) The three-dimensional structure of rocket-exhausted depletions was reconstructed based
- on multi-source observations and simulation.
- (3) The evolution of REDs should be mainly divided into three stages: "rapid formation,
- diffusion-driven growth, and diffusion-driven recovery."

#### 20 Abstract

- The rocket launch process causes a series of disturbances in the ionosphere, among which a
- typical phenomenon is the formation of ionospheric electron density depletions caused by
- chemical reactions involving rocket exhaust, known as Rocket Exhausted Depletions (REDs).
- Current research on the REDs mainly focuses on the horizontal features observed from
- ground-based GNSS data. By utilizing COSMIC radio occultation data, we clearly observed the
- vertical structure of REDs following the launch of an ATLAS-V rocket from Cape Canaveral Air
- Force Station on May 22, 2014. Additionally, combining ground-based GNSS, Swarm satellite
- observations, and numerical simulations, we delineated, for the first time, the three-dimensional
- "hollow tube" structure of the REDs. Then, the spatiotemporal evolution of the REDs is analyzed,
- and considered to mainly consist of three stages: "rapid formation, diffusion-driven growth, and
- diffusion-driven recovery". The study contributes to a deeper understanding of the formation and
- development of artificial ionospheric plasma bubbles.

## 33 Plain Language Summary

- A rocket launch released gases high into the atmosphere and caused a large region where the
- number of free electrons dropped sharply. We combined satellite measurements, ground
- observations, and computer simulations to reveal the three-dimensional shape and evolution of
- this electron loss for the first time. The depletion formed quickly, expanded as the gases spread,

- and then slowly recovered. These results help us understand how frequent launches briefly disturb
- the space environment above Earth.

#### 1 Introduction

- During rocket launches, the ionosphere undergoes a range of physical and chemical interactions,
- resulting in various disturbances. Rockets traveling at supersonic speeds through the mesosphere,
- alongside the explosive release of exhaust gases, generate shock waves and Atmospheric
- Acoustic-Gravity Waves (AGWs) (e.g., Arendt, 1971; Noble, 1990; Jacobson & Carlos, 1994; Li et
- al., 1994). These disturbances induce traveling ionospheric disturbances (TIDs), which are
- frequently observed within a certain range along the rocket's trajectory and have been extensively
- documented (e.g., Kakinami et al., 2013; Lin et al., 2017; Chou et al., 2018; Yasyukevich et al.,
- 2024). The rocket's propulsion relies on rapidly ejecting large volumes of combustion products,
- part of which are released into the atmosphere and can impact climate and stratospheric ozone
- (Barker et al., 2024). As the rocket ascends into the ionosphere, these exhaust gases expand
- rapidly in the ionosphere, they act like a "snowplow", pushing background plasma outward and
- forming a density pile-up layer (e.g., Booker et al., 1961; Mendillo et al., 1988). Simultaneously,
- the exhaust pressure drops sharply to match the background ionosphere, transitioning into a free
- diffusion process. This exhaust, rich in H2O, H2, and CO2, undergoes a series of chemical
- reactions during diffusion, further depleting ionospheric electrons and forming ionospheric
- electron density "holes" known as rocket-exhausted depletions (REDs) (e.g., Mendillo et al., 1975,
- 2008; Bernhardt et al., 1961, 2001). These chemical reactions produce excited oxygen atoms
- (O(1D)) and hydroxyl radicals (OH), which emit 630 nm red light and 135.6 nm ultraviolet
- emissions, observable by airglow imagers, sounding rockets, and satellite instruments (e.g.,
- Mendillo et al., 2008; Liu et al., 2006; Park et al., 2022).
- REDs were first detected by sounding instruments (Booker, 1961) and through Faraday rotation of
- satellite signals (Mendillo et al., 1975; Wand and Mendillo, 1984). Subsequent studies primarily
- relied on GNSS-TEC data to capture the 2D horizontal distribution of REDs (e.g., Mendillo et al.,
- 2008; Furuya and Heki, 2008; Nakashima and Heki, 2014). Furthermore, Park et al. (2015, 2016,
- 2022) observed associated electron density depletions using in-situ satellite measurements even
- six hours after the rocket launch, and captured 2D depletion distributions using satellite ultraviolet
- spectrometers. Numerous observations indicate that REDs typically emerge around 5-7 minutes
- after launch, persisting for 0.5 to 6 hours (e.g., Bernhardt et al., 2001; Mendillo et al., 2008;
- Nakashima & Heki, 2014; Park et al., 2016). The depletion regions generally extend laterally
- along the rocket's trajectory, with widths of approximately 300-500 km and lengths exceeding
- 2000 km (e.g., Liu et al., 2018; Ozeki & Heki, 2010; Mendillo et al., 2008; Zhao et al., 2024).
- Inside REDs, total electron content (TEC) drops by 3-22 TECU compared to the background
- ionosphere (e.g., Liu et al., 2018; Park et al., 2022; Zhao et al., 2024), while maximum electron
- density reductions range from 20% to 95% (e.g., Furuya & Heki, 2008; Park et al., 2016; Mendillo
- et al., 1984, 2008; Zhao et al., 2024).
- Current REDs observations largely rely on GNSS-TEC data, with a few nighttime launches
- observable through optical imaging (e.g., Mendillo et al., 2008; Park et al., 2022), which mainly
- capture horizontal 2D structures. Vertical structure observations remain scarce. A limited number
- of studies using incoherent scatter radar (e.g., Wand & Mendillo, 1984; Bernhardt et al., 1998,

- 2012; Zhao et al., 2024) have captured vertical profiles of REDs. Zhao et al. (2024) reported
- observations of ionospheric REDs structure during two rocket launches, and the results showed
- that REDs can extend to ~200-700 km in altitude. The maximum depletion altitude for the
- afternoon event is 425 km, and the maximum depletion altitude for the midnight event is 283 km.
- Park et al. (2015, 2016) detected REDs diffusing to satellite orbit heights (450 km and 518 km)
- through Swarm satellite in-situ measurements. Park et al. (2022) also utilized the GOLD imager,
- Madrigal TEC, and multiple Low-Earth-Orbit satellites, with COSMIC-2 data revealing an
- increase in ionospheric slab thickness at the depletion center, indirectly supporting vertical
- structure analysis. However, the 3D structure of rocket-exhausted electron density depletion
- remains unclear.
- In this study, we obtained clear observations of the vertical structure of REDs using COSMIC-1
- occultation data. Combining multi-source observations from COSMIC-1, Swarm satellites, and
- ground-based GNSS, we revealed the 3D hollow-tube structure of REDs and their evolutionary
- characteristics, offering a new perspective on rocket launch ionospheric disturbances. Additionally,
- high-resolution 3D simulations further validated the feasibility and reliability of the depletion
- modeling.

#### 96 2 Data and Simulation

#### 2.1 Rocket Launch Event

- Event-1: On May 22, 2014, an Atlas-V Rocket was launched from Cape Canaveral Air Force
- Station by United Launch Alliance (ULA) at 13:09 UT. Event-2: On May 20, 2015, a similar
- Atlas-V was launched at the same station by ULA at 15:09 UT. REDs from two launch events
- were reported by Park et al. (2016) using satellite in-situ observations. This study primarily
- focuses on Event-1, while Event-2 serves as a supplementary case to provide horizontal
- observational data where Event-1 lacks coverage. Detailed trajectories of these two launches were
- not available; therefore, we derived the trajectories based on the rocket depletion observations by
- Park et al. (2015, 2016) and the Atlas-V user manual, as depicted in Figure 1c. The launch
- information and Atlas-V user manual can be accessed from the ULA website:
- https://www.ulalaunch.com/missions.

#### 108 2.2 Data and Methods

- The Constellation Observing System for Meteorology, Ionosphere, and Climate (COSMIC) radio
- occultation data have been widely applied in studies of the atmosphere, climate, and ionosphere.
- The gridded data products derived from COSMIC occultation observations have been used in
- rocket-induced depletion (RED) studies (Park et al., 2022). Moreover, COSMIC occultation data
- are frequently employed to monitor and investigate ionospheric disturbances caused by special
- events such as earthquakes, tsunamis, and sporadic Es layers (Astafyeva et al., 2011; Arras &
- Wickert, 2017; Yan et al., 2018, 2020, 2022; Qiu et al., 2021). A detailed assessment of the
- feasibility and reliability of COSMIC occultation data can be found in Yan et al. (2022). In this
- study, we utilize the electron density and total electron content (TEC) data derived from
- COSMIC-1 occultations (Level 1b, 1/60 Hz; product identifiers: ionPhs\_repro and podTec\_repro)
  to identify electron density depletions induced by rocket exhausts and to determine their
- corresponding altitude information. The COSMIC-1 datasets are provided by the COSMIC Data

Analysis and Archive Center (CDAAC) and are available at https://data.cosmic.ucar.edu/gnss-ro/.

- To investigate the horizontal spatial distribution of the REDs, we used data from ground-based
- GNSS receivers. The vertical total electron content (TEC) was calculated following the method
- described by Yan et al. (2017). The GNSS receiver data were obtained from the SOPAC & CSRC
- database service website (http://garner.ucsd.edu/). The ionospheric total electron content (TEC),
- defined as the total number of electrons integrated along the signal path I (unit:  $m^{-2}$ ,  $10^{16}$   $m^{-2} = 1$
- TECU), was derived from dual-frequency GPS carrier phase (L1/L2) and pseudorange (P1/P2)
- measurements. The computation was based on the ionospheric refraction model proposed by
- Klobuchar (1991):

$$STEC_{L} = \left[ \left( \frac{f_{2}^{2}}{f_{1}^{2} - f_{2}^{2}} \right) \frac{f_{1}^{2}}{40.3} \right] (L_{1}\lambda_{1} - L_{2}\lambda_{2})$$
 (1)

$$STEC_{p} = \left[ \left( \frac{f_{2}^{2}}{f_{1}^{2} - f_{2}^{2}} \right) \frac{f_{1}^{2}}{40.3} \right] (L_{1}\lambda_{1} - L_{2}\lambda_{2})$$
 (2)

$$STEC = STEC_P + \sqrt{\sum_{i=1}^{N} (STEC_L - STEC_P)^2/N})$$
 (3)

- where  $f_1$  and  $f_2$  are GPS signal frequencies at 1.57542 GHz and 1.2276 GHz, respectively;  $\lambda_1$  and
- $\lambda_2$  are the corresponding wavelengths; N is the number of measurements sampled during a satellite
- pass.
- The calculated STEC was projected onto the sub-ionospheric point (SIP) on the Earth's surface
- using an ionospheric single-layer model. The vertical TEC (VTEC) can then be derived from the
- following equation (Jin et al., 2008):

$$VTEC = (STEC - B^S - B_R) \times \sqrt{1 - \left(\frac{r_e cos\theta}{r_e + h_{ion}}\right)^2}$$
 (4)

- where  $B^S$  and  $B_R$  are the instrumental biases related to GPS satellites and receivers, respectively;  $r_e$
- = 6371 km is the mean radius of the Earth; h is the elevation angle of a GPS satellite; hion is the
- height of the single ionosphere model, 350 km in this study. Below we use the TEC in place of
- VTEC for convenience.
- Based on the TEC data, the identification method proposed by Pradipta et al. (2015) for
- ionospheric plasma bubbles was adopted and optimized using a third-order polynomial fitting to
- capture the specific characteristics of REDs. This approach allows for a more accurate extraction
- of the absolute vertical TEC depletion values and provides a clearer representation of the
- horizontal distribution features of the REDs.

- The Swarm constellation, comprising three satellites at orbital altitudes of 450-550 km, is
- equipped with Langmuir probes to measure electron density, enabling the observation of REDs.
- Park et al. (2016) first reported Swarm observations of REDs, with detailed descriptions of the
- instruments and data available in their study. The Swarm data are sourced from the European
- Space Agency (ESA): https://swarm-diss.eo.esa.int/#swarm.

Figure 1. Observation Data Distribution and Details. (a) Map of rocket launch data distribution: the orange line represents the rocket trajectory, the red triangle marks the launch site, the red "×" indicates the start and end points of the rocket's second-stage engine working; The projection of COSMIC observation location points are shown as blue and purple lines, with triangular markers indicating detected depletion locations. The red/green/blue symbols represent REDs center observed by Swarm-Alpha/Bravo/Charlie, the gray dashed lines their trajectories. (b) 3D rocket trajectory (geographic projections) with COSMIC puncture-line shading indicating depletion extents. (c) Electron density measurements from Swarm satellites, with UT time at each marked point. (d) COSMIC TEC profile, with shaded REDs region and corresponding UT time. (e) COSMIC electron density profile.

#### 2.2 Simulation of RED

The formation of ionospheric electron density depletion is closely linked to the diffusion patterns 168 and chemical reactions of rocket-exhaust (e.g. Bowden et al., 2020; Zhao et al., 2024). Previous studies have shown that releasing around 4 kg of water vapor at 210 km altitude can cause 169 170 significant depletions, with the affected area expanding at higher altitudes (e.g. Hu et al., 2010, 171 2011, 2013; Huang et al., 2011). Factors like release trajectory, exhaust flow rate, source speed, 172 geomagnetic declination, and background winds further influence depletion patterns (e.g. Zhao et 173 al., 2016; Feng et al., 2017; Gao et al., 2021). In this study, we also incorporate the effects of 174 daytime electric fields and photoionization to improve simulation accuracy. 175 Rocket launches consume up to 79% of their fuel below 80 km altitude, representing the 176 predominant portion of total fuel consumption (Barker et al., 2024). The ATLAS-V first-stage core carries 284 tons of fuel, with boosters attached to it, while the second stage carries 20.83 tons. The 177 178 first stage alone accounts for over 93% of the total fuel load. As it releases most exhaust in the 179 lower atmosphere and below the ionospheric D-region, the influence of first-stage exhaust on the 180 REDs analyzed in this study is negligible. We selected the second stage ignition (~260 seconds 181 after launch at ~200 km altitude) as the starting point of a 684-second exhaust release. The 182 second-stage engine thrust for the ATLAS-V is approximately 22,890 lbs (equivalent to 10,382.73 kg), with a specific impulse  $(I_{sp})$  of 449.7 seconds. Based on the standard formula relating thrust, 183 184 specific impulse, and exhaust flow rate(Feng et al., 2021):

$$I_{sp} = \frac{F}{g_{o,in}} \tag{5}$$

where *I<sub>sp</sub>* is the specific impulse, *F* represents thrust, and *m* is the released flow rate. The mass flow rate is calculated to be approximately 23.08 kg/s. Assuming a 5.5:1 oxidizer-to-fuel ratio, the mass fraction of water vapor in the exhaust is estimated to be ~95%, and hydrogen ~5%(Mendillo et al., 1975).

The diffusion equation proposed by Bernhardt (1976) for neutral material release calculates the

191 molecular density of a point source as:

$$\begin{split} n(x,y,z,t) &= \frac{N_0}{(4\pi D_0 t)^{1.5}} \exp\{-(z-z_0) \left(\frac{3}{4H_a} + \frac{1}{2H_r}\right) \\ &- \frac{H_a^2 \{-(z-z_0)/(2H_a)\}^2}{D_0 t} - \beta t \\ &- \frac{(x^2+y^2) \exp[-(z-z_0)/(2H_a)]}{4D_0 t} \\ &- \left(\frac{1}{H_a} - \frac{1}{H_r}\right)^2 \frac{D_0 t \exp[(z-z_0)/(2H_a)]}{4} \} \end{split}$$

 $D_0$  is the diffusion coefficient follows Mendillo (1993);  $H_a$  and  $H_r$  (H=kT/mg) are the atmospheric scale heights for air and the release substance, respectively;  $z_0$  is the release altitude; where  $\beta$  is a loss coefficient that includes chemical reactions and photoionization. By moving point sources along the rocket trajectory, we simulate continuous rocket exhaust diffusion (e.g. Zhao et al., 2016; Feng et al., 2021).  $H_2O$  and  $H_2$  released into the ionosphere mainly participates in the following reactions as table-1, to deplete ionospheric electrons:

Table-1 The main chemical equations involved in the simulation release.

| Species        | Reaction Equation                                                                                   | Reaction Ratio cm <sup>-3</sup> ·s <sup>-1</sup> | Reference                |  |
|----------------|-----------------------------------------------------------------------------------------------------|--------------------------------------------------|--------------------------|--|
| H <sub>2</sub> | $H_2 + O^+ \xrightarrow{k_1} OH^+ + H + 0.35eV$                                                     | $k_1 = 1.7 \times 10^{-9}$                       | Ferguson ., 1973         |  |
|                | $OH^++e^- \xrightarrow{k_2} O^* + H+8.74eV$                                                         | $k_2 = 7.5 \times 10^{-8} [300/T_e]^{0.5}$       | Bernhardt ., 1987        |  |
|                | $OH^{+} + H_{2} \xrightarrow{k_{3}} H_{2}O^{+} + H + 1.21eV$                                        | $k_3 = 1.5 \times 10^{-9}$                       | Fehsenfeld et al., 1967  |  |
| $H_2O$         | $H_2O+O^+ \xrightarrow{k_4} H_2O^+ + O+1.01eV$                                                      | $k_4 = 3.2 \times 10^{-9}$                       | Smith et al., 1978       |  |
|                | $H_2O^+ + e^- \xrightarrow{k_5} OH^* + H + 7.45eV$                                                  | $k_5 = 6.5 \times 10^{-7} [300/T_e]^{0.5}$       | Bernhardt ., 1978        |  |
|                | $H_2O+H_2O^+ \xrightarrow{k_6} H_3O^+ + OH+1.17eV$                                                  | $k_6 = 1.67 \times 10^{-9}$                      | Bolden and Twiddy., 1972 |  |
|                | $H_3O^+ + e^- \xrightarrow{k_7} \begin{cases} H_2O + H + 6.29eV \\ OH^* + H_2 + 5.63eV \end{cases}$ | $k_7 = 6.3 \times 10^{-7} [300/T_e]^{0.5}$       | Heppner et al., 1976     |  |

- k1 k7 represent chemical reaction rates;  $T_e$  is electron temperature. Based on the methodology of
- Mendillo et al. (1993), the electron density variation per time step  $\Delta t$  resulting from chemical
- reactions of the type  $A+B \xrightarrow{k_i} C+D$  (where  $k_i$  represents the reaction rate coefficient) is calculated.
- The expression for the concentration changes of reactants and products involved in the reaction
- 203 per time step  $\Delta t$  is given as:

$$\Delta n_i = k_i \cdot n_A \cdot n_B \cdot \Delta t \tag{7}$$

- Neutral release disrupts ionospheric equilibrium, inducing plasma diffusion. Assuming the
- dominant influence of the geomagnetic field, plasma diffusion is primarily motion along magnetic
- field lines, with its continuity equation expressed as:

$$\frac{\partial n_p}{\partial t} = -\nabla \bullet (n_p \vec{v}_\perp + n_p \vec{v}_\parallel) + P_p - L_p \tag{8}$$

- Where  $n_p$  is charged particle density distribution,  $P_p$  and  $L_p$  are particle production and loss terms;
- $208 \quad v_{\parallel}$  and  $v_{\perp}$  are the velocity vector parallel and perpendicular to the magnetic field. Geomagnetic
- inclination (I) and declination (φ) define the field direction, with s along the magnetic field line.
- Plasma diffusion speed along the magnetic field is expressed as:

$$v_{\parallel} = -D_{p} \left[ \frac{\partial \ln(n_{p} T_{p})}{\partial s} + \frac{\sin I}{H_{p}} \right] + v_{D}$$
(9)

$$v_{\perp} = \frac{\vec{E} \times \vec{B}}{B^2} + \frac{m}{q} \frac{\vec{g} \times \vec{B}}{B^2} \tag{10}$$

- $D_p$  is the plasma diffusion coefficient,  $D_p = (1 + T_e/T_i)D_i$ , where  $D_i$  is the ion diffusion coefficient;
- $T_p$  is plasma temperature,  $T_p = (T_i + T_e)/2$ ;  $H_p$  is the plasma scale height,  $H_p = k(T_e + T_i)/m_i g$ ;  $v_D$
- represents external drift velocity;  $\vec{g}$  is gravitational acceleration, and q is the ion charge. The
- $\vec{E} \times \vec{B}$  drift velocity term for  $v_{\perp}$  is provided by Anderson (1978). Based on these equations, the
- plasma diffusion formula in a Cartesian coordinate system (x-east, y-north, z-up) is:

$$\begin{split} \frac{\partial n_p}{\partial t} &= \sin^2 I \cdot \left[ \frac{\partial D_p}{\partial z} \frac{\partial n_p}{\partial z} + \frac{n_p}{T_p} \frac{\partial D_p}{\partial z} \frac{\partial T_p}{\partial z} + \frac{n_p}{H_p} \frac{\partial D_p}{\partial z} + \frac{D_p}{T_p} \frac{\partial n_p}{\partial z} \frac{\partial T_p}{\partial z} + \frac{D_p}{H_p} \frac{\partial n_p}{\partial z} - v_D \sin I \frac{\partial n_p}{\partial z} + \frac{D_p}{H_p} \frac{\partial n_p}{\partial z} + \frac{D_p}{H_p} \frac{\partial n_p}{\partial z} - v_D \cos I \sin \varphi \frac{\partial n_p}{\partial z} + \frac{D_p}{H_p} \frac{\partial n_p}{\partial z} + \frac{D_p}{H_p} \frac{\partial n_p}{\partial z} - v_D \cos I \cos \varphi \frac{\partial n_p}{\partial z} + \frac{D_p}{H_p} \frac{\partial n_p}{\partial z} + \frac{D_p}{H_p} \frac{\partial n_p}{\partial z} - \frac{D_p}{H_p} \frac{\partial n_p}{\partial z} - \frac{D_p}{H_p} \frac{\partial n_p}{\partial z} + \frac{D_p}{H_p} \frac{\partial n_p}{\partial z} +$$

- The numerical model, built upon the above theoretical framework, simulates the launch scenario
- of Event 1 using a central finite-difference scheme. Table 2 summarizes the background models
- and simulation parameters.

Table 2 Parameter settings and background model for numerical simulation.

| Num | parameter                              | value                                                                       |
|-----|----------------------------------------|-----------------------------------------------------------------------------|
| 1   | Release time                           | UT 2014-May-22 13:09                                                        |
| 2   | Location and gird count                | 31.03°N to 19.29°N, 49.80°W to 77.01°W;<br>Long grids=300; Lat grids = 130; |
| 3   | Altitude                               | 100-600  km; $dz = 2  km$ , $Grids = 250$                                   |
| 4   | Time step                              | 0.01s                                                                       |
| 5   | Rate of release                        | H <sub>2</sub> O: 21.9722kg/s & H <sub>2</sub> : 1.1078kg/s                 |
| 6   | Speed of release position              | 3.1-5.9km/s                                                                 |
| 7   | Background ionosphere                  | IRI-2016                                                                    |
| 8   | Background magnetic field              | IGRF-13                                                                     |
| 9   | Atmosphere density and gas temperature | ATMOSNRLMSISE-00                                                            |

## 220 3 Result and Discussion

#### 221 3.1 Observation

- Figure 1(a,b) presents the rocket trajectory and observed depletion locations for Event-1. Figure
- 1(d,e) presents the vertical profiles of TEC and electron density from COSMIC-1 satellite C01
- paired with navigation satellites G01 and G11, labeled as C01-G11 and C01-G01. Both
- occultation events occurred within 10 minutes, with triangular markers indicating the depletion
- centers and corresponding UT timestamps. For the C01-G11 at UT 13:58, about 40 minutes after
- the ATLAS-V rocket's second-stage ignition, depletion was observed between 197-300 km altitude,
- showing a maximum TEC drop of ~50 TECU and an electron density reduction of ~75% at 250
- 229 km. The C01-G01 at UT 13:53 recorded depletion between 310-400 km, with an electron density
- ~15% reduction at 375 km. The lower boundary, estimated using Pradipta et al. (2015), was
- between 250-320 km for C01-G01. Because the vertical profiles of occultation data not only
- 251 between 250-520 km for Cor-Got. Because the vertical profiles of occuration data not only
- record altitude information but also extend along the north-south direction across several thousand
- kilometers, an occultation ray may intersect the horizontal extent of the REDs region. Therefore,
- the upper and lower boundaries observed in occultation profiles do not necessarily represent the
- true vertical limits of the REDs.
- As supporting evidence, these REDs were previously reported by Park et al. (2015, 2016) using
- Swarm in-situ measurements, which are also retrieved in this study (Figure 1c).
- Swarm-Alpha/Bravo/Charlie are marked in different colors, with times of minimum depletion

https://doi.org/10.5194/egusphere-2025-5515 Preprint. Discussion started: 19 November 2025 © Author(s) 2025. CC BY 4.0 License.

266

labeled. The Swarm observations occurred around UT 14:00. Altitudes are shown in Figure 1c, 240 where Swarm A and C orbited at 469.9 km, and Swarm B at 518.5 km. The north-south REDs lengths recorded by Swarm A, C, and B were 419 km, 491 km, and 277 km, respectively, with 241 242 maximum electron density reductions of about 45%, 39%, and 5%. The horizontal distribution of 243 REDs measured by Swarm closely followed the rocket's projected ground track. 244 Figure 2 presents the REDs characteristics extracted from ground-based GNSS TEC data for Event-1 and Event-2. Figure 2a shows the GNSS data distribution for Event-1, Figure 2b shows 245 246 the corresponding time series of the identified REDs in differential TEC (DTEC). The vertical axis 247 is arranged according to the closest distance from the observation points to the rocket trajectory, 248 indicating the observed REDs' distance from the rocket path. Due to the offshore launch and 249 limited temporal coverage, most GNSS data failed to capture the REDs. Some ionospheric 250 piercing points (IPPs) near the second-stage ignition site and occultation region didn't capture 251 RED signatures because they arrived 2-3 hours after launch, by which time the RED had drifted 252 northward out of the area. To improve visualization, the data points without detected REDs 253 signatures are semi-hidden. Figure 2a also shows the horizontal TEC distribution map, where red 254 small triangles indicate the recording position for each data curve at the launch time, and black 255 arrows mark the time sequence of data measurement. For Event-1, the GNSS effective observation data is sparse; the maximum depletion amplitude observed occurred approximately 1.5 hours after 256 257 the launch, with a magnitude of about 9 TECU (1TECU=10^16/m2) at a distance of ~ 200 km 258 from the rocket trajectory, located near the C01-G01. 259 Due to sparse GNSS coverage along Event-1's rocket trajectory, accurate horizontal depletion 260 scales could not be determined. Therefore, Event-2 serves as a reference, given the identical rocket 261 model, similar launch trajectory, and matching local time. Figures 2c-f depict the horizontal TEC 262 distribution and time-series depletion signals from Event-2. Figures 2c and 2e show the IPPs for 263 TEC observed at 300 km by the G14 and G31. The corresponding time-series signals in Figure 2d 264 and Figure 2f are arranged by the shortest distance to the rocket trajectory, similar to Figure 2b. 265 For Event-2, the REDs extended up to ~500 km across the trajectory and exceeded 2000 km in

length, with a maximum TEC reduction of approximately 20 TECU, lasting over 2 hours.

Figure 2. GNSS TEC data for Event-1 and Event-2. (a, c, e) TEC IPPs maps: the large red triangle marks the launch site, the orange line shows the rocket trajectory, small red triangles indicate IPPs at launch time, black arrows show their movement, and color represents depletion magnitude. (b, d, f) Time series of extracted TEC depletion; red dashed lines mark launch time. (a-b) Event-1: (a) IPPs distribution; (b) Depletion time series. (c-f) Event-2: (c, e) IPPs from G14 and G31; (d, f) Corresponding depletion time series.

#### **3.2 Simulation**

Figure 3. Simulation molecular density distribution of H2O and electrons. (a-d) H2O molecular density distribution along the rocket trajectory (height vs. flight distance). (e-h) Electron density distributions along the rocket trajectory. (a-h) Time (lower-right corners): minutes after second-stage ignition; Red line is rocket trajectory; Red star is current release position.

Figure 3 shows the simulation result of H2O release and electron density depletion for Event1. Figure 3 (a-d) illustrates the evolution of H2O molecular density along the rocket flight path. Water vapor diffuses rapidly within the first few minutes, spreading laterally along the trajectory, with a vertical range of around 100 km. Molecular density decreases gradually as diffusion slows, reaching maximum spread at about 25 minutes, before gravitational settling pulls it to lower altitudes, ceasing its contribution to depletion consistent with Zhao et al. (2024). Figure 3 (e-h) shows electron density changes during water release. Depletion forms within 1-2 minutes, spreading for about 20 minutes. And depletion mainly extends laterally along the trajectory, reaching 180 km upwards and 50 km downwards. Recovery follows, with faster recovery at lower altitudes due to higher background density, showing an upward drift pattern consistent with Zhao et al. (2024). At 50 minutes, the depletion reaches a thickness of 150-300 km and 200-500 km in vertical range. We have included multiple detailed depletion evolution videos in the supplementary video materials (sv1.mp4, sv2.mp4, sv3.mp4), which visualize the simulated 3D spatiotemporal evolution processes.

## 3.3 The evaluation of ELE Hole

Figure 4. TEC signals of the depletion observed within the REDs at several different time intervals. (a) Map of piercing points: red triangles indicate the positions of piercing points at the rocket launch time; small red squares mark the positions where the maximum TEC values of the REDs were recorded; blue circles represent GPS stations; red pentagrams indicate the rocket launch site; red dashed lines show the central line of the REDs. (b) Extracted TEC profiles of the depletion: red dashed lines denote the rocket launch time; red triangles and squares in different shades correspond to the elements shown in the left panel.

The ionospheric piercing points (IPPs) for TEC data derived from ground-based GPS receivers shift over time, providing temporal sequences of REDs observations at different intervals. As shown in Figure 4, for the IPPs closest to the rocket trajectory (blue line), the TEC begins to drop approximately 10 minutes after launch—this 10 mins delay corresponds to the time required for the rocket to reach the vicinity of this IPPs. The decrease occurs rapidly within 3-5 minutes, followed by a slower decline. Since this location (blue line) is at the edge of the main REDs, the maximum depletion amplitude observed here is about 10 TECU, which is weaker than the values recorded later by IPPs that pass directly through the depletion (green and pink lines). For IPPs entering the REDs more than 20 minutes after launch, the TEC variations exhibit smoother curves. The maximum depletion amplitude for this entire event is recorded around 90 minutes post-launch. Furthermore, as indicated in Figure 2, most TEC variations observed after 90 minutes are collectively weaker than the maximum amplitude captured by the green line. This suggests that the REDs subsequently entered a diffusion-driven recovery stage after 90 minutes, gradually returning to background levels.

The horizontal evolution of REDs in Event-1 is consistent with Event-2 (same rocket type) and prior cases (e.g. Mendillo et al.,2008), where depletions exceeded >2000 km in length and ~500 km in width from the same launch site. The REDs evolution generally follows three stages: (i) formation within 5-7 minutes post-launch, with GNSS detecting rapid depletion; (ii) diffusion-driven growth over 25-30 minutes, expanding to 500-2200 km in length and 300-500 km in width; (iii) gradual recovery lasting over 50 minutes as density returns to background levels (e.g., Liu et al., 2018; Ozeki & Heki, 2010; Mendillo et al., 1976, 2008; Zhao et al., 2024). Vertical structures were observed by Wand et al. (1984) between 200-500 km in a trajectory matching Event-1. Zhao et al. (2024) reported vertical ranges of ~200-700 km and ~202-535 km

from two rocket launches, classifying the vertical evolution into generation, diffusion, and recovery stages-consistent with horizontal evolution patterns.

Simulations show water vapor rapidly diffuses within 1-2 minutes, expand more slowly over ~20 minutes, and eventually settle into lower atmospheric layers due to gravity; This aligns with previous studies (Hu et al., 2010, 2011; Gao et al., 2017; Zhao et al., 2024). The resulting REDs exhibit a vertically "top-wide, bottom-narrow" profile, consistent with Gao et al. (2017). This structure likely results from: (i) an exponential increase in diffusion coefficients with altitude, causing wider upper depletion; (ii) higher lower-altitude electron densities, promoting faster recovery and forming a narrower base. This matches Wand et al. (1984) and aligns with COSMIC-1 occultation results, confirming the asymmetric vertical structure. GNSS data show TEC drops within 5-7 minutes of launch, lasting 1-2 minutes, and diffusing over 500 km in 15-25 minutes, consistent with simulations (Mendillo, 1976, 2008; Heki & Nakashima, 2010; Bernhardt, 2008). Simulation began 260 s post-launch (second-stage ignition), showing rapid density drops within 2 minutes and peak diffusion at 20 minutes, closely matching GNSS observations. The simulated vertical range (200-500 km) matches Swarm-A/C observations at 469 km and COSMIC-1 events, while Swarm-B at 518 km, near the edge, recorded only ~5% variation, within simulation error margins.

#### 3.4 The structure of ELE Hole

Figure 5. Simulated electron density distribution at 40-minute. (a) The COSMIC-1 data tangent points (marked by dashed lines) with electron density profile projections on the 40°N vertical cross-section. The other marks are the same as Figure 1. (b-f) The simulated and observed ranges in different latitudinal ranges; (g-k) The percentage change in electron density.

Figure 1 shows COSMIC-1 occultation data and rocket data. Previous studies (Park et al., 2015, 2016) reported similar REDs lasting nearly 6 hours using DMSP satellite data. For Event-1, GNSS data measured depletions ~5 TECU even 3.5 hours post-launch, indicating a prolonged lifetime. Both COSMIC-1 occultations occurred within 50 minutes post-launch, observing depletions

within 300 km of the rocket trajectory, confirming those depletions originated from the launch. The C01-G11 showed >75% electron density reduction, while C01-G01 recorded a smaller reduction <20%. Swarm-A and C, observing along the latter half of the trajectory, detected reductions around 40%. Time differences between these four datasets were <30 minutes. The location of maximum depletion for C01-G11 was ~110 km horizontally from the trajectory, while that for C01-G01 exceeded 300 km. This suggests that depletion strength depends on occultation proximity to the center. Similarly, Swarm-B at 519 km altitude, near the depletion's edge, recorded ~5% variation, while Swarm-A and C at 469 km, closer to the center, observed larger reductions. Multi-distance observations aided in positioning the 3D spatial structure of the REDs. However, the occultation tangent point reflects both vertical and horizontal variations, so it doesn't directly indicate depletion thickness. Since the occultation ray path is aligned east-west, matching the rocket trajectory, the associated uncertainty has minimal impact on the RED reconstruction. Therefore, the east-west systematic error in occultation observations can be neglected when estimating the 3D hollow-tube structure.

Table 3. The Characteristics of RED by Rocket Launch Events in previous studies; TSLC is Taiyuan Satellite Launch Center of China (38.5°N, 111.6°E), KSC is Kennedy Space Center of USA(28.5°N, 80.7°W).

| Num | Rocket<br>Type | Launch<br>Station | Launch Time (UT) |       | RED Scale(km) | RED           | _                     |
|-----|----------------|-------------------|------------------|-------|---------------|---------------|-----------------------|
|     |                |                   | Date             | Time  | Length*Width  | life<br>(min) | Reference             |
| E-1 | LM-4B          | TSLC              | 2013 Dec 9       | 03:26 | 1300*450      | ~120          | Liu et al., 2018      |
| E-2 | LM-4B          | TSLC              | 2014 Dec 7       | 03:26 | 1300*450      | ~120          |                       |
| E-3 | LM-2D          | TSLC              | 2023 Mar 30      | 10:50 | >1500*~150    |               | Xie et al., 2025      |
| E-4 | LM-6A          | TSLC              | 2023 Sep 10      | 04:30 | 2000*~300     |               |                       |
| E-5 | LM-6A          | TSLC              | 2022 Mar 29      | 09:05 | 2600*~300     | ~126          | Zhao et al., 2024     |
| E-6 | Titan IV       | KSC               | 2005 Apr 30      | 00:50 | 2200*520      |               | Mendillo et al., 2008 |
| E-7 | Taepodong-1    | North Korea       | 1998 Aug 31      | 02:30 | *~200         |               | Ozeki & Heki, 2010    |
| E-8 | Taepodong-2    | North Korea       | 2009 Apr 5       | 03:07 | *280          |               |                       |

The observational period of the radio occultation data falls within the first 50 minutes after the rocket launch. Relative to the RED's total lifetime of nearly six hours, this observational period places the RED in the early stage of its diffusion recovery phase, a period characterized by a considerable horizontal extent. According to previous studies summarized in Table 3, the horizontal width of the RED ranges from 150 to 520 km, depending on factors such as rocket type and launch trajectory. The RED generated by rockets of the same type performing the same orbital mission exhibit similar widths. Therefore, the REDs produced by Event-1 and Event-2 can be assumed to have comparable widths, approximately 500 km, which is used as the horizontal constraint for the three-dimensional RED structure.

Figure 5 illustrates the simulated electron density distribution and the estimated spatial extent of the REDs constrained by multi-dimensional observational data. The horizontal extent of the REDs is constrained by GNSS TEC observations, while the vertical scale is jointly constrained by COSMIC and Swarm satellite altitude observations. The REDs boundary calculated by numerical simulation is approximated as an ellipse-like shape. Based on the horizontal and vertical constraints, the ellipse-like boundaries are fitted to the COSMIC radio occultation and Swarm

satellite observations. Polynomial fitting is then used to connect these boundaries, producing the 385 red dashed tubular volume in Figure 5, representing the observed REDs extent. The 386 three-dimensional simulated electron density distribution is displayed in cross-sectional view as 387 six parallel colored panels in Figure 5. By comparing the simulated RED with the fitted 388 observational RED, the horizontal distance between the centers of the simulated and observed 389 REDs does not exceed 120 km, and for the COSMIC-1 C01-G11 occultation event, the distance 390 between the observed and simulated RED centers is less than 30 km. The simulated RED is about 391 20% wider than the observed electron depletion, likely due to discrepancies between the 392 background parameters used in the simulation and the actual ionospheric conditions. Differences 393 between IRI model outputs and real ionospheric TEC can exceed 20% (He et al., 2023). As the 394 RED extends over thousands of kilometers, it is subjected to varying background wind intensities 395 and associated uncertainties at different locations. These variations, in turn, induce differential 396 RED drift distances, resulting in positional discrepancies between the simulated RED centerline 397 and the observed 3D centerline. 398 Combining simulation and observational results, the three-dimensional structure of the REDs is a 399 "flattened hollow tube" with a wider top. The entire "flattened 3D hollow tube" envelops the 400 launch trajectory, with the trajectory located closer to the lower side of the tube, resulting in an 401 asymmetric vertical distribution of the RED along the trajectory. Simulation results indicate that 402 the width and thickness of the "flattened 3D hollow tube" are primarily controlled by the amount 403 and altitude of exhaust release and the background electron density distribution. The High-density 404 area electron density layer is mainly located around 300 km in the ionospheric F1 layer, where 405 most chemical reactions of the rocket exhaust also occur, positioning the electron density 406 depletion center near the F1 layer. As atmospheric density decreases with altitude, the diffusion 407 coefficient of the exhaust increases exponentially with height (Mendillo, 1975), causing upward 408 diffusion to be faster and forming an upper-wide, lower-narrow, quasi-ellipsoidal shape. Exhaust 409 released at lower altitudes can also diffuse upward toward the F1 layer, producing stronger 410 chemical reactions and generating larger amplitude electron density depletions at higher altitudes 411 along the release trajectory.

## 4 Conclusion

- This study first utilizes COSMIC-1 occultation data to resolve the vertical structure of REDs,
- integrates Swarm and GNSS-TEC observations, and reconstructs its 3D hollow-tube morphology.
- Observation-simulation comparisons validate model reliability and support a three-stage RED
- evolution framework: rapid formation, diffusion-driven growth, and recovery. The main
- conclusions are as follows:
- 1. Using COSMIC-1 radio occultation data, we for the first time observed the vertical distribution
- of REDs at different locations along the rocket trajectory. At a location 700 km from the launch
- site, the RED vertical extent is 197-300 km, while at another location 2600 km away, the observed
- vertical extent is 310-400 km.
- 2. By combining multi-dimensional observational data with three-dimensional numerical
- simulations, we reconstructed the three-dimensional tubular structure of the RED. Its vertical
- cross-section is an upper-wide, lower-narrow quasi-elliptical shape, with a vertical-to-horizontal
- thickness-to-width ratio of approximately 1:2. The horizontal width of the RED is mainly

- controlled by the amount and altitude of rocket exhaust release.
- 3. Based on observations and simulations, the evolution of rocket-exhausted ionospheric electron
- density depletion can be divided into three stages: rapid formation, diffusive growth, and diffusive
- recovery. During the first 3-5 minutes after exhaust release, the REDs undergoes rapid growth,
- with the fastest rate of electron density decrease. It then enters a 15-30 minute diffusive growth
- stage, during which the REDs expands to its maximum spatial extent, with a vertical thickness
- ranging from 100-500 km and a horizontal width along the trajectory of 300-500 km. Finally, the
- REDs evolves into the diffusive recovery stage, lasting more than 50 minutes, longer than the
- preceding two stages. During this stage, the REDs slowly returns to background values while
- undergoing drift motions at different rates due to the influence of the magnetic field and
- background wind.
- Due to the influence of factors such as local time, propellant characteristics, and orbital insertion,
- understanding of REDs' 3D evolution remains limited. Broader observational coverage and more
- diverse cases are needed to identify common patterns and better assess REDs' physical drivers and
- space weather impacts.

#### 441 Data availability

- All the data can be provided by the corresponding author upon request. And the observational data
- in this study can be found at: (a) GNSS: http://garner.ucsd.edu/pub/rinex/, (b) Swarm:
- https://swarm-diss.eo.esa.int/#swarm. (c) COSMIC-1/pod: https://data.cosmic.ucar.edu/gnss-ro/co
- smic1/repro2021/level1b/.

#### 446 Acknowledgments

- This work is supported by the National Natural Science Foundation of China (NSFC, 42074191).
- We gratefully acknowledge the Scripps Orbit and Permanent Array Center (SOPAC) and the
- California Spatial Reference Center (CSRC) for providing publicly accessible GNSS data
- (http://garner.ucsd.edu/pub/rinex/). We also acknowledge the University Corporation for
- Atmospheric Research (UCAR) for providing COSMIC-1 radio occultation data via the COSMIC
- Data Analysis and Archive Center (CDAAC) at https://data.cosmic.ucar.edu/. Swarm satellite data
- were obtained from the European Space Agency's Earth Online Swarm Dissemination Server
- (https://swarm-diss.eo.esa.int/), and we thank ESA for making these data available.

## 455 Competing interests

- The contact author has declared that neither they nor their co-authors have any competing
- interests.

## **Author contributions**

- XY conceptualized this study. CD carried out the data analysis and numerical simulations with
- comments from other co-authors. CD and XY wrote the original manuscript. CX, TY, and YQ

- provided data processing and constructive suggestions. All authors contributed to the improvement
- of the paper.

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
