# Peer review of "Three-Dimensional Hollow Tubular Structure of Rocket"

_EGUsphere, 2025_

## Referee Comment (RC1)

Review of Three-Dimensional Hollow Tubular 1 Structure of Rocket Chemical Depletion by Chunyu Deng, Xiangxiang Yan, Tao Yu, Chunliang Xia, and Yifan Qi.

This paper has merit for publication but some wording should be changed. The following items should be considered:

- (0) Line 23: The term "Rocket Exhausted Depletions (REDs)" is awkward and ambiguous. A better term would be "Holes in the Ionosphere from Rocket Exhaust (HIREs)"
- (1) Line 32: The statement "The study contributes to a deeper understanding of the formation and development of artificial ionospheric plasma bubbles." is misleading. The word bubbles refers to equatorial structures that rise in altitude. The plasma depletions produce by rocket exhaust do not "bubble" up but just stay at a fixed altitude. Please replace "bubbles" by "hollow-tubes".
- (2) Line 57 has "Bernhardt et al., 1961, 2001". Bernhardt was not writing papers in 1961. Please use the correct date and that check this paper is in the references.
- (3) Line 413 has "This study first utilizes COSMIC-1 occultation data to resolve the vertical structure of REDs, integrates Swarm and GNSS-TEC observations, and reconstructs its 3D hollow-tube morphology." should be replaced with "This is the first study that utilizes COSMIC-1 occultation data to resolve the vertical structure of REDs, integrates Swarm and GNSS-TEC observations, and reconstructs its 3D hollow-tube morphology."

---

## Author Comment (AC1)

We sincerely appreciate and are honored to receive your insightful comments, which have significantly improved the quality and clarity of our manuscript. We have diligently addressed all the raised concerns and our detailed responses are presented as follows:

(0) Line 23: The term "Rocket Exhausted Depletions (REDs)" is awkward and ambiguous. A better term would be "Holes in the Ionosphere from Rocket Exhaust (HIREs)"

Response:We agree that the term "Rocket Exhausted Depletions (REDs)" is awkward and ambiguous. Accordingly, we have replaced it throughout the manuscript with the clearer and more descriptive term "Holes in the Ionosphere from Rocket Exhaust (HIREs)."

(1) Line 32: The statement "The study contributes to a deeper understanding of the formation and development of artificial ionospheric plasma bubbles." is misleading. The word bubbles refers to equatorial structures that rise in altitude. The plasma depletions produce by rocket exhaust do not "bubble" up but just stay at a fixed altitude. Please replace "bubbles" by "hollow-tubes".

Response: We appreciate this important clarification. To avoid confusion between the term "bubbles" and equatorial plasma bubbles, and to better reflect the structures observed in this study, we have replaced "bubbles" with "hollow-tubes" in the revised manuscript.

(2) Line 57 has "Bernhardt et al., 1961, 2001". Bernhardt was not writing papers in 1961. Please use the correct date and that check this paper is in the references.

Response: We sincerely apologize for the citation error. The reference to 1961 was mistakenly attributed to Bernhardt; the correct 1961 reference is Booker and is already cited elsewhere in the manuscript. We have corrected "Bernhardt et al., 1961, 2001" to "Bernhardt et al., 2001" in Line 57. During this revision, we also rechecked the references and corrected the other inconsistency.

(3) Line 413 has "This study first utilizes COSMIC-1 occultation data to resolve the vertical structure of REDs, integrates Swarm and GNSS-TEC observations, and reconstructs its 3D hollow-tube morphology." should be replaced with "This is the first study that utilizes COSMIC-1 occultation data to resolve the vertical structure of REDs, integrates Swarm and GNSS-TEC observations, and reconstructs its 3D hollowtube morphology."

Response: Thank you for helping us improve the clarity of our wording. The sentence in Line 413 has been revised as suggested: "This is the first study that utilizes COSMIC-1 occultation data to resolve the vertical structure of HIREs, integrates Swa rm and GNSS-TEC observations, and reconstructs their 3D hollow-tube morphology."

Once again, we sincerely thank the reviewer for the careful reading and constructive feedback.